# The Divergent Pluripotent States in Mouse and Human Cells

**DOI:** 10.3390/genes13081459

**Published:** 2022-08-16

**Authors:** Xuepeng Wang, Qiang Wu

**Affiliations:** The State Key Laboratory of Quality Research in Chinese Medicine, Macau 999078, China

**Keywords:** early embryo development, pluripotent stem cells (PSCs), naive pluripotency, primed pluripotency, formative pluripotency, signaling pathways, gene regulatory network, epigenetic modifications

## Abstract

Pluripotent stem cells (PSCs), which can self-renew and give rise to all cell types in all three germ layers, have great potential in regenerative medicine. Recent studies have shown that PSCs can have three distinct but interrelated pluripotent states: naive, formative, and primed. The PSCs of each state are derived from different stages of the early developing embryo and can be maintained in culture by different molecular mechanisms. In this review, we summarize the current understanding on features of the three pluripotent states and review the underlying molecular mechanisms of maintaining their identities. Lastly, we discuss the interrelation and transition among these pluripotency states. We believe that comprehending the divergence of pluripotent states is essential to fully harness the great potential of stem cells in regenerative medicine.

## 1. Introduction

In early mammalian embryonic development, from zygote to two-cell-stage blastomeres (in mice) or to 8-cell-stage blastomeres (in humans) are considered totipotent, which is the ability to give rise to an embryo and all its supportive extra-embryonic tissues [1,2,3]. Along blastomeres being fated to inner cell mass (ICM) or trophectoderm (TE), ICM is further committed to epiblast and hypoblast (also named primitive endoderm). The epiblast, the source of pluripotent stem cells (PSCs), keeps developmental potential to give rise to three germ layers tissues. However, they are incapable of extra-embryonic linage differentiation. The cultured PSCs can be divided into two classes according to their potency. First-class PSCs can differentiate into both embryonic and extra-embryonic cell lineages. The second-class PSCs, derived from the epiblast, can only differentiate into embryonic lineages. Interestingly, a low-population cluster in mouse PSCs can be found to share features with the two-cell-stage mouse embryos in transcriptome, epigenome, and developmental potential. Thus, they are named two-cell-like cells (2CLCs) [1]. Recently, human 8C-like cells (8CLCs), which could produce embryonic and extra-embryonic lineages, were identified [2]. Furthermore, extended pluripotent stem cells (EPSCs) with both embryonic and extra-embryonic potency were established from human and mouse pluripotent stem cells [4]. In addition, the so-called expanded potential stem cells (EPSCs) from mouse eight-cell-stage blastomeres were also established. These EPSCs are enriched with molecular signatures of blastomeres and possess developmental potency for all embryonic and extra-embryonic cell lineages [5]. Notably, porcine EPSCs and human EPSCs were also successfully established [6].

Second-class pluripotent stem cells derived from epiblasts show three states: naive, formative, and primed; although these cells are capable of differentiation into three germ layers, they exhibit many differences in cell morphology, signaling requirement, epigenetic modifications, metabolome, and transcription profiling [7,8].

Researchers have dedicated significant effort to capturing PSCs at different stages in culture dishes and characterizing them. In this endeavor, high-throughput sequencing, single-cell analysis, and other advanced technologies have been especially useful in assessing the fidelity of in-vitro PSCs to their in-vivo counterparts [9,10].

The three states of pluripotent stem cells, the naive, formative, and primed states, can be obtained from both mouse and human embryos or through epigenetic resetting and reprogramming [8,9,11,12,13,14,15,16]. Mouse embryonic stem cells (mESCs) can self-renew indefinitely in vitro while preserving the developmental potential to reconstitute all embryonic cell types. mESCs are considered to be in the naive state since they are derived from pre-implantation embryos. They are characterized by high pluripotency and can contribute to chimeras and X activation in females [7]. Chimeras refer to an organism comprising at least two populations of genetically distinct cells. They are an invaluable tool for studying mammalian development and PSCs’ potential [17,18]. Naive mESCs can be reintroduced into host embryos to contribute to the mouse development. Moreover, the naive pluripotent stem cells must undergo a period of conversion (called capacitation) and enter the early post-implantation epiblast-like state before they are enabled to respond to cues for trilineage germ-layer differentiation [19,20]. Epiblast stem cells (EpiSCs) are derived from the columnar epithelial epiblast of the post-implantation embryo. They are defined as a primed state, with random X inactivation in females and cannot form chimera [12,21]. Between preimplantation and the epithelialized stage of the egg cylinder of late post-implantation, other intermediate stem cell states are mostly heterogeneous and have not been fully characterized [9,22]. These intermediate stem cells are referred to as formative pluripotency, corresponding to the cells of the early post-implantation epiblast [8]. Formative stem cells can rapidly respond to the induction of lineage specification (especially the germline lineage), cues for cell polarity, and do not show lineage priming [8]. There have been many efforts to capture stem cells in their formative state in vitro. Recently, a population of intermediate stem cells from E5.5-E6 mouse embryos was captured as formative stem (FS) cells. Interestingly, FS cells can be directly induced into primordial germ cells (PGCs) and form chimera at lower efficiency [13]. Similarly, two other groups reported their success in the derivation of formative cell lines (namely fPSCs and XPSCs). These cells share common features with FS cells, including PGCs induction and transcription profiling [15,16]. Additionally, rosette-like stem cells (RSCs) derived from epiblast RSCs express naive markers KLF4, NANOG, and ESRRB, while upregulating *Otx2*, *Tcf7l1*, *Podxl,* and *Cgn* and repressing primed marker *OCT6* [14].

It is well known that integrated cooperation among signals, transcription factors (TF), and epigenetic modifications is required for the maintenance of pluripotent stem cells [7,12,13,15,16,21]. Hence, uncovering the molecular mechanisms underlying pluripotency states is a pre-requisite to stem cell therapy and regenerative medicine. Further, stem cells in different pluripotent states can be an excellent model to study embryo development as naive, formative, and primed cells perfectly mimic embryos at pre-implantation, early post-implantation, and late post-implantation stages.

In this review, we discuss the underlying mechanisms of maintenance and inter-conversion among different states of pluripotent stem cells. We focus on our current knowledge of the signaling pathways, transcriptional networks, and epigenetic modifications that contribute to sustaining various pluripotent states of murine ESCs. We surmise that this hierarchy of pluripotent states may be extended to in-vitro totipotency. Thus, one can expect a clear regulatory pathway starting from totipotency to primed pluripotency, and finally, to commitment.

## 2. The Characteristics of Naive, Formative, and Primed Stem Cells

In mammals, naive, formative, and primed pluripotent stem cells can be established from early embryos at distinct developmental stages, corresponding to their gradually restricted pluripotency [7,12,13,21,23]. mESCs were firstly established from the ICM or the epiblast from pre-implantation blastocysts [24,25]. ESCs have higher pluripotency compared to formative and primed stem cells, as shown by the efficient chimera formation [26]. Pluripotent stem cells can also be derived from the columnar epithelial epiblast of the post-implantation embryos. These so-called EpiSCs also have potential to differentiate into multilineage cell types [12,21]. Both ESCs and EpiSCs express key pluripotency genes *Oct4/Sox2/Nanog* that are crucial for pluripotency maintenance. However, there are significant differences between these two stem cell types, including cell morphology, X-chromosome state, essential signaling pathways, and chimeric formation ability. In comparison with the dome morphology of naive ESCs, EpiSCs have flatter colony morphology. In addition, female EpiSCs are X inactivated (XaXi), whereas X chromosomes are activated (XaXa) in female naive ESCs. Furthermore, maintenance of EpiSCs in culture dishes requires FGF and activin A, while naive ESCs need leukemia inhibitory factor (LIF) and would be driven into differentiation by FGF/Erk signaling [12,21,27,28]. Strikingly, EpiSCs are unable to integrate into host blastocysts [12,21]. These suggest that EpiSCs have restricted pluripotency in line with in-vivo embryo development and are ready to differentiate. Therefore, EpiSCs pluripotent stem cells are in a primed state.

In mouse embryo development, the blastocyst is formed at about E3.5 days, in which its outer layer is trophoblasts and its inner layer is ICM [29]. As development progresses to E4.5, ICM develops into epiblast and primitive endoderm (also known as hypoblast). As the embryo implants, naive pluripotency transcription factors decrease and their ability to form ESCs is lost, while transcription factors, such as Otx2 and Pou3f1, are up-regulated [23,26,29,30]. At about the E6.5 stage, a population of cells in the epiblast can be induced into PGCs. Interestingly, naive ESCs are unresponsive to germ cell inductive stimuli directly unless they transit into epiblast-like cells (EpiLCs), which the ESCs exit from naive pluripotency and establish a new gene regulatory network. Furthermore, cells that have lost ES cell identity do not immediately upregulate definitive lineage specification markers [31,32,33]. Thus, only EpiSCs are refractory to PGC induction [34,35,36]. Naturally, researchers have hypothesized that the third pluripotent state stem cells, ‘formative’ stem cells, can be directly induced into PGC production. They may have features that are different from naive and primed stem cells. During the formative stage, cells may switch from an apolar state to a polarized epithelial state. Meanwhile, naive transcription factors, such as Nanog, Esrrb, and Klf4, are shut off and formative genes are activated instead [37]. Three laboratories have successfully captured epiblast-like cells that possess formative state characters. These cells are called formative stem (FS) cells, formative pluripotent stem cells (fPSCs), and XPSCs, respectively [13,15,16]. FS cells, fPSCs, and XPSCs can respond directly to the stimulation of germ cell induction and be induced into PGCs. Morphologically, these formative stem cells (FSCs) show polarized epithelioid and rosette structures, like the recently reported rosette stem cells (RSCs) [14]. At the transcriptome level, formative stem cells have up-regulated formative genes *Otx2*, *Fgf5,* and *Oct6* and down-regulated naive marker genes *Esrrb*, *Tcf2l1*, and *Klf4*. Like naive ESCs and primed EpiSCs, formative stem cells can self-renew and differentiate into all three layers. Furthermore, formative stem cells have higher pluripotency than EpiSCs because they can form chimeras, although the efficiency is lower than that of naive ESCs [13,15,16].

The restricted potency from ESCs to FSCs and to EpiSCs appears to mirror the developmental progression from the naive preimplantation epiblast to the post-implantation epiblast to the epithelialized egg cylinder [8,33,38]. Whole-transcription analysis of these three states of pluripotent stem cells shows that naive ESCs are closer to pre-implantation epiblast (E4.5), formative stem cells are related to pre-gastrulae cells (E5.5–E6.5) while EpiSCs are similar with mid-gastrulae cells (E7.0) [15]. Consistently, the Gene Ontology (GO) term enrichment analysis highlighted ‘cell adhesion’, ‘gastrulation, and development’ in FS cells and EpiSCs, respectively. Single-cell sequencing indicates that formative stem cells similar to pre-/early-gastrula epiblasts and 48h EpiLCs differ from naive ESCs and primed EpiSCs [13,15,39]. Further analysis shows that mouse XPSCs and RSCs are closer to ∼E5.0 epiblasts, while mouse FS cells and fPSCs are closer to ∼E5.5–E6.5 epiblasts and the initiation point of gastrulation [9]. Consistently, XPSCs and RSCs express higher levels of naive pluripotency genes, such as *Klf4,* compared to FS cells and fPSCs. Female XPSCs and RSCs also exhibit two X chromosome activations (XaXa), like naive ESCs, while female FS cells and fPSCs exhibit one X chromosome inactivation (XaXi) [37]. Taken together, these results indicate that naive ESCs are unprepared to execute lineage decisions and must undergo a maturation process. Primed EpiSCs have initiated a response to inductive cues and are already partially specified and fate biased. Formative stem cells are between naive and primed pluripotency and are poised to quickly respond to patterning and lineage specification cues. We summarize the features of the naive, formative, and primed PSCs in mice and humans (Figure 1).

## 3. Signaling Pathways for Naive, Formative, and Primed State of Stem Cells

Naive mESCs are first cultured on mitotically inactivated mouse embryonic fibroblast (MEF) with fetal calf serum (MEF/serum condition) [24,25]. Leukemia inhibitory factor (LIF) secreted by MEF is essential to support self-renewal of ESC culture (LIF/serum condition). LIF binds to LIFR and gp130, and activates transcription factor Stat3, thus, prompting target gene *Esrrb*, *Klf4*, *Nanog*, *c-Myc*, and *Tfp2l1* expression to inhibit ESC differentiation and promote viability [40,41,42,43]. LIF ligands can also activate the PI3K/AKT and MAPK signaling pathways, which are dispensable for the maintenance of ESCs since blocking these pathways does not affect ESC pluripotency [40,42,43,44]. However, without serum, LIF alone is not sufficient to sustain ESCs from differentiation toward neurectoderm. Bone morphogenetic proteins (BMPs) were found to be the primary component in serum to inhibit neural ectoderm differentiation in ESCs. BMP4 plus an LIF culture condition can support long-term self-renewal in mouse ESCs [45].

BMP signaling has a crucial function in mESC maintenance. BMP4 activates SMAD1/5, and phosphorylated Smad1/5 forms a complex with Smad4 to enter the nucleus to initiate *Id* (inhibitor of differentiation) expression. BMP4/Smad1/5 can also suppress Erk5 activity. In addition, BMP4 can crosstalk with FGF/Erk to suppress ESC differentiation. Interestingly, blocking BMP4/SMAD1/5 signaling pathway by knockout *Smad1/5* does not affect the naive state. However, blocking BMP/Erk5 signaling by BMP4 antagonist noggin or Erk5 activator will cause dramatic ESC differentiation [46,47,48]. Thus, BMP4 maintains the naive state of ESCs mainly by inhibiting Erk5 activity. In addition, BMP4 is required for the generation of primordial germ cells in the mouse embryo [49]. Consistently, the PGC precursors emerge from the most proximal layer of the epiblast, where the BMP–Smad signaling is strongly activated. In the Bmp4-heterozygous mutants, the number of PGCs is almost halved, which is also the case in the double heterozygous Smad1 and Smad5 [50].

FGF/Erk is the primary signaling pathway causing differentiation of naive ESCs. FGF receptor inhibitor or downstream MEK inhibitor PD0325901 can prevent ESC differentiation, even without LIF and BMP4 [51,52]. In addition, the GSK inhibitor CHIR99021 can enhance ESC derivation in a highly efficient manner [53]. This culture condition is called “2i”. 2i plus LIF (2i/LIF) can enhance the maintenance of stable naive ESCs. More importantly, the 2i/LIF culture condition applies not only to other mouse ESCs, but also to rat ES cell line establishment [51], suggesting that these signaling pathways are critical and evolutionarily conserved.

Maintaining formative PSCs relies on distinct extrinsic signaling pathways. Formative stem cells can be captured with different signaling combinations. Using WNT signaling pathway inhibitor XAV939 and low-dose activin A (3 ng/mL), FS cells can be successfully isolated from E5.5–E6 mouse embryos and D5-D6 human embryos. Furthermore, this culture condition can be applied to induce FS cells from naive ESCs in both mice and humans [13]. Interestingly, Wu and colleagues derived formative-like PSCs and named these cells after FGF, TGF-β/activin A, and Wnt activation (FTW). The FTW conditions were also applicable to horse and human cells and the resultant formative stem cells showed high potential for interspecies chimerism [16]. Compared with the 2D culture system, 3D culture offers the environment to facilitate cell–cell and cell–matrix interactions and complex transport dynamics for nutrients; thus, 3D culture is more like the in-vivo condition [54]. Using a 3D culture system (Matrigel as scaffold), Wang et al. captured formative stem cells (fPSCs) by inhibiting the WNT signaling pathway and turning on the activin A and FGF pathways [15]. RSCs, XPSCs, FS cells, and fPSCs are all PSCs but represent different stages of the formative epiblast. Mouse XPSCs and RSCs are closer to ∼E5.0 epiblasts, while mouse FS cells and fPSCs are closer to ∼E5.5–E6.5 epiblasts and the initiation point of gastrulation [37].

EpiSCs require the FGF/activin A signaling pathways [12,21]. However, murine EpiSCs can also be expanded in an alternative growth condition (GSK3i/ IWR1) without exogenous FGF2/activin A supplementation since IWB1 can function in retention of β-catenin in the cytoplasm [55]. Interestingly, the state of EpiSCs in the GSK3i/IWR1 condition is different from EpiSCs expanded in FGF2/activin A conditions due to the higher levels of expression of naive marker genes. Indeed, murine EpiSCs under the FGF2/IWR1 condition correspond to posterior–proximal epiblasts while the cells in classical FGF2/activin A conditions are close to anterior late-gastrula primitive streak [56].

## 4. Transcriptional Network Governing Pluripotent States

Transcriptional regulators are indispensable for pluripotency maintenance of naive, formative, and primed PSCs [39,57,58,59,60]. These PSCs may be in different states, but a large proportion of pluripotency genes are expressed at similar levels. Such genes include *Oct4*/*Sox2*/*Nanog*, which constitutes the core transcriptional network [9]. However, certain genes are expressed in different levels in these three types of cells. For instance, *Klf4*, *Tfcp2l1*, *Esrrb*, *Klf2*, *Tbx3*, and *Prdm14* are highly expressed in naive ESCs but silenced in formative and primed PSCs [57,58]. The transcription factors encoded by these specific genes constitute a flexible control circuitry that maintains the naive state of the ESCs. Hence, they are known as naive markers [40,61]. Among these TF genes, *Klf4* is obviously down-regulated during the conversion from naive ESCs into primed EpiSCs. Interestingly, Klf4, in combination with Oct4/Sox2/c-Myc, can induce somatic cells into pluripotent stem cells [62]. The importance of Klf4 for naive state maintenance was affirmed by the fact that *Klf4* overexpression in the 2i culture condition was enough to convert primed EpiSCs into naive ESCs [63].

Compared to naive ESCs and primed EpiSCs, formative stem cells possess up-regulated *Otx2, Dnmt3b, Fgf5, Zic2/5, Etv1/4, Oct6,* and *Grhl2*, which were reported to play important roles in the transition from naive to primed state [14,15,64]. Among these genes specific to formative stem cells, *Otx2* is prominently up-regulated during the formative transition [33,65]. It is required to maintain FS cells in a stable state. Knock out *Otx2* in naive ESCs hinders the production of formative stem cells. On the contrary, *Otx2* mutant naive ESCs can be converted into stable primed EpiSCs. Mechanically, Otx2 can redirect the genomic occupancy of Oct4 [66,67]. During the conversion of naive ESCs to formative stem cells, Otx2 plays a leading role in remodeling gene regulatory networks, which recruit Oct4 to many new enhancer regions. Furthermore, Oct4 controls Otx2 level by regulating its transcription and stabilizing its protein. When the cells exit naive pluripotency, increased Otx2 binds to many enhancer regions of the formative genes, which are or are not prebound by Oct4 and activates the correlated genes. Therefore, the Oct4–Otx2 regulatory axis actively establishes a new regulatory chromatin landscape to exit from naive pluripotency and transit into a formative state [67].

EpiSCs are primed pluripotent stem cells with some pre-commitment to lineage specification. They possess down-regulated expression of naive makers, including *Esrrb*, *Klf2* and *Klf4,* and up-regulated lineage genes, such as *Brachyury*, *Foxa2,* and *Zic2* [29]. Similar to formative stem cells, primed EpiSCs preferentially utilize the proximal enhancer but not the distal enhancer of Oct4.

In summary, the states of pluripotency and inter-conversion are heavily linked with specific gene resetting. As the cells differentiate, naive pluripotency genes are replaced with formative pluripotency genes and primed pluripotency genes. Finally, the stem cells exit from pluripotency and are committed into ectoderm, mesoderm, and endoderm lineages cells.

## 5. Pluripotent-State-Associated Epigenetic Modifications

In mammalian development, epigenetic modifications, including DNA methylation and histone modifications, play a critical role in conferring the pluripotency of ESCs [68]. The conversion of naive ESCs into formative stem cells or primed EpiSCs requires epigenetic remodeling [13,15,69,70,71]. Firstly, DNA is hypomethylated in naive ESCs while DNA is hypermethylated when ESCs differentiate into FSCs or EpiSCs. DNA hypomethylation usually represents the active chromatin state. Similar to naive stem cells, two-cell-like cells exhibited lower global DNA methylation [72]. Secondly, in naive female ESCs and female XPSCs and RSCs (close to ∼E5.0 epiblast), both X chromosomes are activated (XaXa). In contrast, in female FS cells and fPSCs (close to ∼E5.5–E6.5), one of the X chromosomes is inactivated along with the increased levels of histone H3K27me3. Consistently, the chimerism efficiencies of XPSCs and RSCs are notably higher than those of FS cells and fPSCs, but lower than those of naive ESCs. Thirdly, more bivalent histone modifications are harbored in the developmental genes in formative stem cells and primed EpiSCs than naive ESCs [73,74,75]. These suggest that primed stem cells are more prone to respond to differentiation cues.

In addition, noncoding RNAs (ncRNAs) have been found to be involved in the conversion of these three pluripotent states, especially Xist, a well-known long noncoding RNA (lncRNA) [76]. During the naive ESC transition into formative or primed PSCs, female X-chromosome inactivation (XCI) will occur. The completion of XCI firstly needs Xist RNA to coat the X chromosome [77]. Afterwards, PRC1/2 is recruited to exclude RNA pol II and active histone modifications and the addition of repression histone marks, H3K27me and H3K9me2 [78]. Thus, Xist RNA is the initial step of XCI and crucial for the maintenance of pluripotent state.

Epigenetic features are important to identify naive ESCs, especially in human ESCs, because chimera experiments cannot be used to evaluate naive hESCs [79,80,81]. Notably, HDACi can reset human-primed EpiSCs into naive ESCs, emphasizing the importance of the epigenome [11]. Furthermore, recently, 8C-like cells (8CLCs) were established by the addition of TSA (histone deacetylase inhibitor) and DZNep (inhibitor of histone H3K27 methyltransferase EZH2). This evidence highlights the importance of chromatin conformation changes in rolling human pluripotent stem cells back to earlier developmental stages [2]. However, epigenetic changes are usually subtle and not sufficient to clearly distinguish divergent pluripotent states.

## 6. Discussion

Pluripotent stem cells are derived from the epiblast at the preimplantation stage to the mid-gastrulation stage during early embryo development [82]. Capturing pluripotent stem cells can not only provide excellent platforms to study early embryonic development and cell fate conversions for regenerative medicine applications. Naive ESCs derived from the ICM of a blastocyst are not biased to differentiate and can produce chimeras. These ESCs require minimal extrinsic signaling and can be stably cultured under the 2i/LIF condition or any combination of these three factors (2i + LIF, 2i, LIF + CH, LIF + PD) [7,58,61]. Furthermore, ESCs can efficiently transit from one culture condition to another without affecting pluripotency or cell survival [58]. Though the chromatin state of naive ESCs is less restricted and more open, naive ESCs cannot differentiate directly into PGCs [31]. It is noteworthy that genomic instability and retrotransposon activity is elevated while telomere maintenance declines during the conversion of naive ESCs to primed EpiSCs [83]. This suggests that the compromised potency in EpiSCs is determined by the changes in chromatin states.

Formative stem cells possess an intermediate state of pluripotency between the naive and the primed PSCs [13,15]. PGCs and lineage differentiation can be more quickly induced from formative stem cells than from naive ESCs. Formative stem cells require FGF and activin A activation but not the LIF/2i condition. They can generate chimeras at lower efficiencies and can differentiate into PGCs, showing that formative stem cells are beginning to exit from full pluripotency. EpiSCs can neither form chimeras nor produce PGCs, showing that EpiSCs derived from post-implantation egg cylinder represent fate-biased pluripotency [7].

Notably, the conversion among diverse pluripotent states can be achieved in vitro. Consistent with embryo development direction, naive ESCs can be easily induced into formative stem cells or EpiSCs and formative stem cells also easily transit to EpiSCs [13]. However, reversion-primed EpiSCs back into naive ESCs requires a combination of Klf4 over-expression with the 2i/LIF condition [63]. Whether over-expression of certain pluripotent-state-specific genes can be universally utilized to reprogram more primed stem cells back into a naiver state remains unknown and is worth trying.

It is noteworthy that researchers also put efforts into capturing PSCs in intermediate states between naive pluripotency and totipotency. Further, 2C-like cells that express totipotency genes *Zscan4s* and *Mervl* are the first reported cells capable of producing intra- and extra-embryonic tissues, although they are not stable in the medium and only exist in a very low percentage [1,3]. In 2017, two research groups reported the establishment of extended/expanded pluripotent stem cells (EPSCs) using different combinations of small molecules [4,5]. The extended pluripotent stem cells capable of generating both embryonic and extraembryonic lineages from both humans and mice can be captured by using an LCSD chemical cocktail (LIF, CHIR99021, (S)-(+)-dimethindene maleate (DiM) and minocycline hydrochloride (MiH). These mouse EPSCs show robust and superior chimeric ability at the single-cell level, while the human EPSCs show interspecies chimeric competency in mouse conceptuses [4]. The EPLSCs can also be established by the cocktail (LIF, CHIR99021, PD0325901, JNK Inhibitor VIII, SB203580, A-419259, and XAV939), which can modulate key developmental pathways. Importantly, the EPSCs hold bidirectional differentiation potential in both intra- and extra-embryonic differentiation [5]. Furthermore, totipotent blastomere-like cells (TBLCs), which are close to 2-cell and 4-cell stage embryos, can be established in vitro by inhibiting the spliceosome [84]. The less compromised pluripotency also provides a good model for research and applications.

Interestingly, hESCs that are also derived from the pre-implantation epiblast, are considered as being in a primed state since hESCs share similar features with primed mEpiSCs but not mESCs [85,86]. This might be due to species differences between mouse and man. Mouse embryos offer a longer window for isolating naive ESCs [7]. Many efforts have been put into gaining naive hESCs from human embryos or reprograming primed hESCs [19,87,88]. Exogenous KLF2/NANOG transgenes can trigger the conversion from primed to naive pluripotency [11]. Subsequently, several methods have been established to maintain naive hPSCs in vitro using combinations of inhibitors and agonists, such as 2i (MERi; GSKi), HDACi, TGF-β, and FGF agonists [81,87,89,90,91]. In addition, Guo et al. successfully captured human naive pluripotent stem cells directly from isolated cells of the human ICM with a series of kinase inhibitors (PD0325901, CHIR99021, Gö6983, Y-27632 and human LIF) [92]. The cell lines exhibit naive features, such as global DNA hypomethylation, expression of naive pluripotency markers KLF4, TFCP2L1, and DPPA3, active mitochondria, and reduced glucose dependence, and can be propagated by enzymatic dissociation to single cells [92].

## 7. Perspectives

Early embryo development is a fascinating process where the one zygotic cell develops into an entire embryo. Grasping the knowledge of the embryo development is crucial for us to fully understand human health. However, due to the ethics and the scarcity of research materials, scientists cannot directly study human embryo development. Pluripotent stem cells provide an excellent in-vitro model to study early embryo development. The divergent pluripotent stem cells we discussed above perfectly represent different pluripotent stem cells in developmental embryos at different stages. Thus, understanding the differences and transition among these pluripotent stem cells is important in biomedical research.

Since PSCs have the ability that differentiates into specific types of cells under specific stimulation, all pluripotent cells are valuable resources for stem-cell-based therapy and tissue replacement. The global resources of transplantable organs are in short supply. The directed differentiation of PSCs brings hope for future cell therapy and organ transplantation. It is always a challenge to obtain functional and high-purity cells and organs for clinical applications. The divergent pluripotent stem cells provide more options of stem cells resources for cellular differentiation, including extraembryonic cell types, including placenta and umbilical cord cells. In addition, PSCs have always been used for drug screening and disease modeling. Notably, the advanced genome editing technologies can greatly facilitate the utilization of PSCs in genetic modification and gene therapy.

In short, the subtle genetic/epigenetic difference among stem cells in different pluripotent states not only lets us link in-vitro observation with in-vivo developmental processes but also allows us to comprehend the underlying mechanism of maintenance, exit, and acquisition of pluripotency. Such knowledge is invaluable as we seek to manipulate stem cells for regenerative medicine.

## Figures and Tables

**Figure 1 genes-13-01459-f001:**
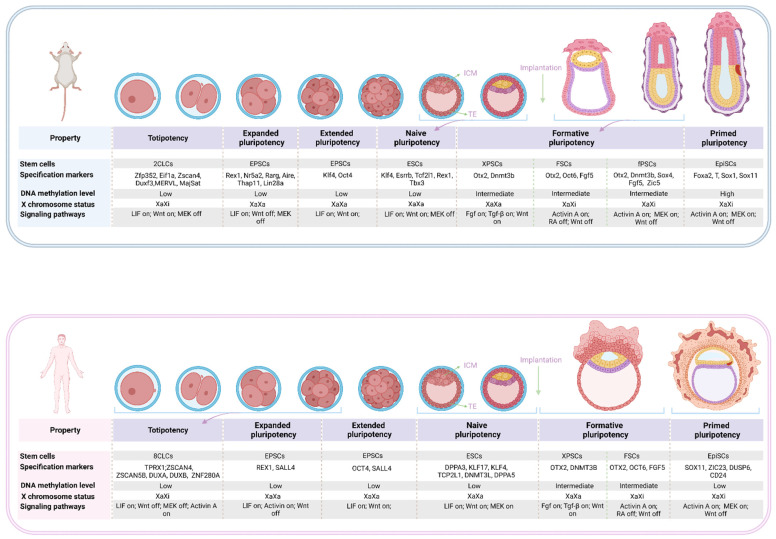
The divergent pluripotent states.

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
