# Peer review of "The Divergent Pluripotent States in Mouse and Human Cells"

_genes, 2022, doi:10.3390/genes13081459_

Round 1

Reviewer 1 Report

Overall, the review article presents interesting information such as the maintenance mechanisms in different stem cell pluripotency states. The figures are suitable, they present a summary of the information in the human species and rodents to make the information more understandable. The knowledge of this information allows to increase what is known about these cells for use in regenerative medicine. As for the references, some from 1993, 1984 could be updated. On page 2, title 2, review the naive term with the punctuation if it is correct or needs to be modified. 

Author Response

We sincerely thank the reviewer for the positive comments and insightful advices. We have accordingly revised our manuscript (in blue color).

Overall, the review article presents interesting information such as the maintenance mechanisms in different stem cell pluripotency states. The figures are suitable, they present a summary of the information on the human species and rodents to make the information more understandable. The knowledge of this information allows to increase what is known about these cells for use in regenerative medicine. As for the references, some from 1993, 1984 could be updated. On page 2, title 2, review the naive term with the punctuation if it is correct or needs to be modified.

Response: Thanks for the advice. we have updated these references and checked all our references carefully. As for the term ‘naive’, we have revised it into the correct form. We also carefully checked the spelling in our manuscript. 

Reviewer 2 Report

The manuscript entitled “The divergent pluripotent states” the authors described the features of the three pluripotent states (naive, formative, and primed), and review the underlying molecular aspects and mechanisms of maintaining their identities in culture.

However, the paper have to be improved with several issues that must be consider.

Major:

1-Lack of detail:

Line52 mESCs detail better characteristics

Line54: Detail better chimera concept with references

Line 72-74: describe better the genes

Line118: detail better

Line 142: detail better

Line197, Ref17: give a brief description of 3D culture system

Line237-239: detail better

Line313: detail better

Line326-329: detail better

Chapter3: Figure summarizing signaling pathways involved

2-Line252: epigenetics also include ncRNAs-> include also examples covering this aspect

3-Line268: highlight the limitations

4-Include a section highlighting the application/implication in development, regenerative medicine and ethical concerns

5-English editing is recommended (i.e.: line45, line108, lines110-112)

Minor:

1-Line52 vs line92: mESCs= state the full name as it’s the first mention

2-Line 105: LIF= state the full name as it’s the first mention

3-Line129: already mention PGCs

4-Line223: TF= state the full name as it’s the first mention

5-Format reference style according to the journal and add when necessary (i.e. lines 131, 136, 159, 205-208-210-213, 246, 273, 287, 316, and all the other parts of the manuscript)

6-Line244: reference missing??

7-Name Figure 1a and b also in the manuscript

Author Response

We sincerely thank the reviewer for the positive comments and insightful advices. We have accordingly revised our manuscript (in blue color).

  • Lack of detail:

Line52 mESCs detail better characteristics

Response: Thanks for the advice. We have supplemented the characteristics of mESCs in lines 51-54.

Line54: Detail better chimera concept with references

Response: Thanks for the advice. We have narrated the concept of chimera and cited two references, shown in lines 55-58.

Line 72-74: describe better the genes

Response: Thanks for the suggestion. We have replenished more genes related to the formation of rosette-like stem cells (lines 77-78)

Line118: detail better

Response: We added more details about how naive ESCs response to differentiation cues.

Line 142: detail better

Response: We have added more detailed information (such as the GO analysis)

about the difference among the naive ESCs, FSCs, and primed EpiSCs (lines 142-149).

Line197, Ref17: give a brief description of 3D culture system

Response: According this advice, we have briefly introduced 3D culture system and cited the related reference (lines 203-205).

Line237-239: detail better

Response: We have accordingly added more details about Otx2 function in formative stem cells (lines 244-247).

Line313: detail better

Response: We added more information of extended/expanded stem cells in our revision (lines 329-337).

Line326-329: detail better

Response: Thanks for the suggestion, we have added more information about human naive ESC establishment (lines 351-355).

Chapter3: Figure summarizing signaling pathways involved

Response: We integrated signaling pathways into Figure 1.

2-Line252: epigenetics also include ncRNAs-> include also examples covering this aspect

Response: Thanks for this advice. We included ncRNAs in our revision (lines 276-283).

3-Line268: highlight the limitations

Response: revised (lines 289-292)

4-Include a section highlighting the application/implication in development, regenerative medicine and ethical concerns

Response: Thanks for this insightful advice. We added our discussion on the topic (lines 357-375).

5-English editing is recommended (i.e.: line45, line108, lines110-112)

 Response: revised.

Minor:

1-       Line52 vs line92: mESCs= state the full name as it’s the first mention

Response: revised

2-Line 105: LIF= state the full name as it’s the first mention

Response: revised

3-Line129: already mention PGCs

Response: revised

4-Line223: TF= state the full name as it’s the first mention

 Response: revised

5-Format reference style according to the journal and add when necessary (i.e. lines 131, 136, 159, 205-208-210-213, 246, 273, 287, 316, and all the other parts of the manuscript)

 Response: revised

6-Line244: reference missing??

 Response: revised

7-Name Figure 1a and b also in the manuscript

Response: revised

Round 2

Reviewer 2 Report

Highlight in the title "in mouse and human cells"